# Bovine Serum Albumin as a Platform for Designing Biologically Active Nanocarriers—Experimental and Computational Studies

**DOI:** 10.3390/ijms25010037

**Published:** 2023-12-19

**Authors:** Olga Adamczyk, Magdalena Szota, Kamil Rakowski, Magdalena Prochownik, Daniel Doveiko, Yu Chen, Barbara Jachimska

**Affiliations:** 1Department of Physics, Cracow University of Technology, 30-084 Krakow, Polandmpr0v98@gmail.com (M.P.); 2Jerzy Haber Institute of Catalysis and Surface Chemistry Polish Academy of Sciences, 31-355 Krakow, Polandkamil.rakowski@ikifp.edu.pl (K.R.); 3Department of Physics, University of Strathclyde, Glasgow G4 0NG, UK; daniel.doveiko.2018@uni.strath.ac.uk (D.D.); y.chen@strath.ac.uk (Y.C.)

**Keywords:** bovine serum albumin, fluorouracil, drug delivery system, ligand–protein interaction, protein structure, molecular docking

## Abstract

Due to the specificity of their structure, protein systems are adapted to carry various ligands. The structure of many proteins potentially allows for two types of immobilization of a therapeutic agent, either on the outer surface of the protein or within the protein structure. The existence of two active sites in BSA’s structure, the so-called Sudlow I and II, was confirmed. The conducted research involved determining the effectiveness of BSA as a potential carrier of 5-fluorouracil (5FU). 5-fluorouracil is a broad-spectrum anticancer drug targeting solid tumors. The research was carried out to estimate the physicochemical properties of the system using complementary measurement techniques. The optimization of the complex formation conditions made it possible to obtain significant correlations between the form of the drug and the effective localization of the active substance in the structure of the protein molecule. The presence of two amino groups in the 5FU structure contributes to the deprotonation of the molecule at high pH values (pH > 8) and the transition to the anionic form (AN1 and AN3). To investigate the binding affinity of the tautomeric form with BSA, UV-vis absorption, fluorescence quenching, zeta potential, QCM-D, and CD spectroscopic studies were performed. The experimental research was supported by molecular dynamics (MD) simulations and molecular docking. The simulations confirm the potential location of 5FU tautomers inside the BSA structure and on its surface.

## 1. Introduction

Drug delivery systems (DDSs) aim to improve the effects of drugs by extending drug release times, improving their biodistribution, increasing the biocompatibility of the carrier, and mitigating the side effects of the treatment [1]. Drug delivery systems take advantage of the unique properties of nanometer-sized particles to improve the biodistribution and pharmacokinetics of active substances [2,3]. Although it has been accepted that nanoparticles should be smaller than 150 nm, it is increasingly reported that due to the discontinuity of the cancer tumor epithelium, the size requirements for nanoparticles developed for cancer treatment are 70 to 200 nm [4]. Nanoparticles, due to their size, are more efficiently internalized into cells than larger microparticles, making them effective transport and delivery systems. The attractiveness of nanoparticles is based on their relatively high ratio of functional surface area to mass, making them able to efficiently bind and carry bioactive compounds [5]. Nano-sized objects are designed at the atomic or molecular level and exhibit unique chemical, magnetic, and biological properties [4,6]. Various nanocarriers, such as liposomes, proteins, dendrimers, polymer nanoparticles, quantum dots, and carbon nanotubes are currently being developed [7]. Regarding enhancing the effectiveness of therapy, stimuli-responsive targeting is often considered an improvement over non-stimuli-responsive systems, which rely on active or passive drug release. In this context, specific physicochemical triggers, such as changes in temperature and pH or the application of electromagnetic waves or ultrasound, are responsible for releasing the pharmaceuticals from the carrier [6,8,9]. While active drug delivery focuses on functionalizing the surface of nanocarriers with ligands designed to target over-expressed receptors on diseased cells, passive drug delivery utilizes the enhanced permeation and retention (EPR) effect. Each method presents its own advantages and challenges, and the choice between them often depends on the carrier type and the active agent involved [7,10]. Biomolecules such as albumins may be promising drug delivery systems due to their preferential uptake by cancer cells or inflamed tissues [11]. Compared to synthetic nanocarriers, proteins are characterized by high biodegradability, a lower cytotoxicity, and immunogenicity [11,12].

Serum albumins are the main component of blood plasma and are responsible for transporting various endogenous and exogenous substances [13]. Research conducted on the mechanism of albumin–drug interactions and the affinity of ligands plays a crucial role in designing systems with the desired therapeutic effectiveness [13]. Bovine serum albumin (BSA) is a protein widely studied as a carrier of active substances, primarily due to its structural similarity to human serum albumin (HSA) [14]. The BSA molecule consists of three domains (I, II, III), each divided into two subdomains (A, B). Its composition contains two tryptophan residues (Trp-134, Trp-213) and 17 disulfide bridges that stabilize the structure [15]. Trp-213 is located in the hydrophobic binding pocket in Sudlow I, while Trp-134 is located on the molecule’s surface [14].

5-Fluorouracil (5FU) has been used to treat a wide range of cancers since 1957, including colorectal cancer and breast cancer [14,15]. This uracil derivative occurs in a neutral form (at pH < 8) or in a deprotonated form (at pH > 8) in aqueous solution [16,17,18]. In the case of the deprotonated form, there are two anionic forms (AN1 and AN3). The form in which the 5FU molecule occurs significantly affects its interactions with proteins and thus, its ability to penetrate cell membranes [18]. The cytotoxicity of 5FU is attributed to its interference with DNA or RNA. In particular, it inhibits the activity of the enzyme thymidylate synthase (TS) [17].

5-Fluorouracil, similarly to other active substances, is transported to tissues in the body via conjugation with serum albumins. The mechanisms of interaction between 5FU and human or bovine serum have been recognized and described in the literature [13,19,20]. Chinnathambi et al., using various types of optical spectroscopy, demonstrated that a single binding site with a high affinity for 5FU characterizes HSA and BSA. In addition, their studies indicate that the albumin–5FU complexes are stabilized by hydrophobic interactions and hydrogen bonds [19,20]. Similar results were obtained by Abdi et al., indicating the presence of hydrogen bonds between oxygen and hydrogen atoms of 5FU and the amino acid residues of the polypeptide backbone of BSA [15].

The available literature data mainly refer to spectroscopic analyses and simulations in determining the mechanism of integrating serum albumin with the neutral form of 5FU. This work focused on a relatively new approach to elucidating the mechanism of BSA complex formation with negatively charged 5FU tautomers. For this purpose, spectroscopic techniques sensitive to changes occurring in the Trp area, such as UV-vis and fluorescence spectroscopy, were used. The spectroscopic studies confirmed the formation of a complex between BSA and the 5FU tautomer and determined the drug’s affinity to the protein. Forming a stable complex did not destructively affect the secondary structure of the protein. A quartz crystal microbalance (QCM-D) was used to assess the changes in physicochemical properties within the protein after the formation of BSA-5FU complexes. This approach, unique in the literature, allows for predicting the potential behavior of the complex when in contact with a negatively charged biological membrane. Changes in the hydrophobicity of the adsorbed layers were verified by contact angle (CA) measurements. Considering the strong influence of carrier surface charge on cellular internalization, the change in zeta potential as a function of pH was determined for the BSA-5FU complexes compared to free albumin. Additional information at the molecular level was obtained through applications of molecular dynamic (MD) simulations and molecular docking. The simulations approximated the distribution of 5FU tautomer molecules both inside the structure and on the surface of the system. Thus, the existence of more active sites than expected for protein systems was proposed.

## 2. Results and Discussion

### 2.1. UV–Visible Spectroscopy Studies

A change in the absorption spectrum of the protein can confirm the complex formation between the protein and ligand. The most significant changes occurred at λ = 279 nm, where absorption comes from the three aromatic amino acids in the protein structure (tryptophan, tyrosine, and phenylalanine) [21,22]. Thus, the effect of ligands on changes in their microenvironment can be observed by UV-vis spectroscopy [21,22]. Figure 1a shows the absorption spectra of native BSA and in the presence of increasing 5FU concentrations (5–30 mg/mL). The absorbance maximum of free protein was concentrated at λ = 279 nm, while the spectra after adding 5FU were blue-shifted to λ = 267 nm (Figure 2b). The observed blue shift may be due to either a decrease in polarity around the BSA chromophores after interaction with 5FU or an increase in the hydrophobicity of the microenvironment around the aromatic amino acid residues, which may be a result of the unfolding of the BSA peptide backbone [20,23].

To further confirm BSA-5FU complex formation, a differential spectrum was plotted by subtracting the spectrum of free 5FU at the same concentration from the spectrum of the BSA-5FU complexes at a molar ratio of 1:20 (Figure 1c). Since the absorption spectrum of 5FU does not overlap with the differential spectrum, a ground-state complex between 5FU and BSA was confirmed [23].

### 2.2. Fluorescence Spectroscopy Studies

Tryptophan is commonly used as a spectroscopic tool to track changes in ligand binding or to determine binding affinity [24]. BSA contains two tryptophan residues: Trp 213, located in a hydrophobic binding pocket in the second domain (Sudlow I), and Trp134 on the BSA surface in the first domain. Tyr residues are distributed in all domains of BSA, equal to 7, 7, and 4 in domains I, II, and III, respectively. In contrast, the HSA structure is characterized by the presence of a single tryptophan residue located in the second domain at position 214 (Trp 214) [14,24]. The intrinsic fluorescence of biomolecules such as albumins can be disrupted by various physicochemical processes, such as ground-state complex formation, energy transfer, exciplex formation, or collisional quenching [19]. To understand the fluorescence quenching mechanism in the BSA-5FU system, we performed BSA titration with negatively charged 5FU. We observed a successive decrease in the fluorescence intensity with increasing drug concentration. Fluorescence quenching of both proteins was studied at wavelengths corresponding to tyrosine and tryptophan (λ_exc_ = 279 nm) and tryptophan alone (λ_exc_ = 296 nm) [25].

The fluorescence spectra of BSA with different concentrations of 5FU were measured and shown in Figure 2. The standard tryptophan emission occurs at λ_em_ = 342 nm. Under the studied conditions (pH = 8.4), the fluorescence emission of albumin (dark blue spectrum) underwent a blue shift to emission λ_em_ = 330 nm, indicating changes in the Trp microenvironment due to conformational modifications in the protein structure as a result of the increasing pH [19]. The fluorescence intensity of albumin successively decreased with the addition of 5FU. Plotting the change of log[(*F*_0_ − *F*)/*F*] versus log[*5FU*] resulting from the quenching of BSA fluorescence derived from tryptophan and tyrosine (λ_ex_ = 279 nm) gave a binding constant of *K_BSA-5FU_* = 1.44 × 10^5^ M^−1^ (Figure 2a). The binding constant at pH 8.4 is, therefore, higher than that at pH 7.5 obtained by Chinnathambi et al. (K = 0.32 × 10^5^ M^−1^) [19], which demonstrated the higher affinity of the 5FU tautomer to albumin compared to neutral drug. Curve fitting to determine the binding constant was not possible after excitation at λ_ex_ = 296 nm when the fluorescence of BSA was dominated by Trp fluorescence (Figure 2b). This may be because the presence of two tryptophan residues in the BSA structure may lead to the overshadowing of the quenching of one of them by 5FU. In this case, changes in the fluorescence intensity of tryptophan were too small. To accurately determine the binding site of 5FU by BSA, the fluorescence quenching of HSA, whose structure is characterized by a single tryptophan residue, was verified. The HSA-5FU binding constant at λ_ex_ = 296 nm was *K_HSA-5FU_* = 1.76 × 10^8^ M^−1^, demonstrating a high 5FU affinity to the tryptophan residue located in the hydrophobic binding pocket in the second domain of HSA (Figure 2c). Comparative fluorescence studies for BSA and HSA indicate a strong effect of the 5FU tautomer on the microenvironment of Trp213 and Trp214, respectively. This indicates that ligands are located close to the Sudlow I binding site, which was further confirmed by the molecular dynamics (MD) analysis.

### 2.3. Determining the Particle Size of BSA-5FU Complexes

The size of the protein and complexes were monitored using dynamic light scattering (DLS). Protein molecules in water at pH 8.4 had a hydrodynamic diameter of 6.3 ± 0.2 nm. This is consistent with literature data, which show that the hydrodynamic radius of BSA, depending on environmental conditions (pH, solvent), is between 3.2 and 4.3 nm [26]. BSA complexes with 5FU had a slightly larger hydrodynamic radius. The average d_H_ value for the complexes before dialysis was 7.4 ± 0.3 nm, and after dialysis, it was 7.3 ± 0.2 nm. The DLS measurements show that the immobilization and association of negatively charged 5FU molecules on the surface of the BSA molecule increases their diameter by approximately 1 nm. For complexes in the range of 1:10–1:60, no protein aggregation was observed. Figure 3 shows the changes in the size of the hydrodynamic diameter for all tested complexes before and after dialysis and shows the size distribution for the 1:20 complex. The PDI value was in the range of PDI = 0.3–0.5, which indicates that the tested system is homogeneous.

### 2.4. Electrophoretic Mobility

Changes in the zeta potential for protein carrier systems in the presence of drugs indicate the immobilization of drug molecules on the albumin surface. The values of the zeta potential as a function of pH for BSA and the BSA-5FU complexes were calculated from the electrophoretic mobility (*μ_e_*) measurements and are presented in Figure 4. The isoelectric point of BSA in an aqueous solution was estimated to be p = 5.1, the same as for albumin dissolved in NaCl with ionic strengths in the range of I = 1 × 10^3^ − 0.15 M [27,28].

BSA’s net charge is positive at pHs lower than 5.1, and for pH values above, i.e., p, it is negative. Above pH = 7.0, the zeta potential value reached a plateau at around ζ = −40.2 ± 1.6 mV. The addition of 5FU to BSA decreased the surface charge of albumin toward more negative values. The most significant change was observed above pH > 7.0, where, relative to BSA, the charge of the complexes dropped by 7% and 20% for excess drugs at ratios of 1:60 and 1:40, respectively. The results indicate that the deprotonated tautomer of 5FU (anionic form) undergoes adsorption on the positively charged amino acids located on the BSA surface, causing compensation of their charge and thus contributing to the amplification of the albumin molecule’s global negative charge. In the case of the 1:40 complex, greater changes in the zeta potential were observed than for the 1:60 complex. This means that too high an addition of the drug causes a decrease in the effectiveness of the complex.

### 2.5. BSA-5FU Adsorption on Negatively Charged Au Surface Monitored by QCM-D

The DLS and zeta potential measurements confirmed the immobilization of drug molecules on the surface of the protein structure. To verify how the presence of drug molecules changes the surface properties of the protein, a series of measurements of the adsorption of BSA/5FU complexes on a gold surface were carried out using the QCM-D method. During the tests, changes in both the resonance frequency and energy dissipation were recorded. The effectiveness of the complexes before dialysis was higher than after dialysis. This can be due to the greater number of drug particles associated with the complex, which eliminates the global charge of the protein. We have a weaker repulsive interaction between protein molecules in the adsorption layer in this situation. After dialysis, the adsorption efficiency of complexes was relatively lower. The highest adsorption efficiency was obtained for the 1:40 complex. These measurements correlate well with the zeta potential results, where the largest changes in zeta potential were obtained for the 1:40 complex.

The changes in Δ*F* and Δ*D* for the measurement series of the complexes after dialysis are presented in Figure 5a,b. Washing the system with a solvent causes minimal desorption, which confirmed that the adsorption process on the gold surface is irreversible, both in the case of BSA and all BSA-5FU complexes. The obtained dissipation energy values for all measurements (Figure 5b) did not exceed the value of Δ*D* = 0.8 × 10^−6^, which indicates that the formed layers are rigid. In the case of frequency changes, the Sauerbrey model was used to obtain the mass adsorbed on the surface. In the case of BSA, the adsorbed mass was Γ_QCM-D_ = 59.5 ng/cm^2^. The adsorbed mass was much higher for all BSA-5FU complexes (Figure 5c,d). For all complexes, the adsorbed mass before dialysis was higher than after dialysis. Still, no linear correlation was observed between the molar ratio of BSA to 5FU and the amount of adsorbed mass. In the case of complexes after dialysis, the adsorbed mass systematically increased up to a molar ratio of 1:40. In the case of 1:60 complexes, a decrease in efficiency was observed. This is consistent with the changes observed in the system’s zeta potential measurements.

Adhesion to cells is essential for carriers used as drug delivery systems. In this context, a critical parameter is the degree of hydrophobicity of the carrier. It is known from the literature data that optimal cell adhesion is observed for systems with a contact angle of 50–70° [29]. For this purpose, the effect of the drug addition on the hydrophobic properties of the protein was verified. The contact angle was measured for the layers adsorbed on the gold surface obtained from the QCM-D measurements. The contact angle for the BSA layer was 67 ± 1°, while for 5FU, it was 56 ± 1°. All layers obtained for the BSA-5FU complexes were in the range of 67–57° (Figure 6). For pre-dialysis systems, the contact angle was higher than for post-dialysis systems. As the adsorbed mass increased, the contact angle decreased. Even though the adsorbed mass was lower, we obtained larger differences for systems after dialysis. In this case, the angle varied from 66 ± 2° to 57 ± 2°. The contact angle for the complex with a BSA-5FU molar ratio of 1:40 was the lowest and similar to that obtained for 5FU. The immobilization of drug molecules on the protein surface affects its hydrophobicity and is in the optimal range for adequate adhesion to cellular systems.

### 2.6. Monitoring Changes in Protein Secondary Structure by Circular Dichroism (CD) Spectroscopy

The structural stability of albumin resulting from its interaction of 5FU was monitored using CD spectroscopy. Figure 7 illustrates the change in CD spectra upon binding of 5FU molecules to the BSA structure in the far UV region. In the BSA spectrum, there is one positive absorption band (maximum) at 193 nm (π–π*) and two negative absorption bands (minimum) at 208 nm (π–π*) and 222 nm (n–π*), which are characteristic of the presence of the α-helical structure [30]. CD spectra were recorded for BSA and protein complexes in ratios from 1:10 to 1:60. The addition of 5FU did not significantly modify the BSA initial spectrum. These studies confirm that the protein conformation remains unchanged after complex formation. Moreover, no changes in the maximum and minimum positions of the spectrum were observed, which may indicate local changes in the structure caused by the drug’s presence in the complex’s structure, both before and after dialysis.

### 2.7. MD Simulation

BSA natively acts as a transport protein and binds several lipophilic compounds in hydrophobic active centers. From a molecular point of view, the localization of ligands that do not belong to the native ligands of a given protein, i.e., are dedicated to a specific structure, may be outside the main active sites. As our previous studies on lactoglobulin showed, they confirmed the immobilization of tetracaine outside the main binding pocket of this protein [31]. Experimental research indicates an identical situation in this case. Therefore, MD simulations and molecular docking were used to confirm this fact.

The isoelectric point of BSA occurs at pH 5.1. At pH 8.4, the protein has a net negative charge [27]. The electrostatic potential distribution determined using APBS shows that the protein surface was predominantly negatively charged. However, the interdomain area was positively charged where the Sudlow I and II active centers are located (Figure 8a,b). Based on the sequence, it is known that the BSA structure has more positively charged residues compared to HSA. The inter-domain space is a good docking site, especially for negatively charged ligands [32].

The 5FU molecule exists in several tautomeric forms. Under physiological conditions, 5FU occurs in a neutral form and as an anion under alkaline conditions. The negative charge is located on nitrogen N1 or N3. According to literature data, the AN1 (DN1) form is more stable than the AN3 (DN3) anion in the gas phase. According to quantum mechanical calculations, the difference in the stability of AN1 and AN3 is approximately 10 kcal/mol. The greater stability of the AN1 anion results from the more favorable delocalization of the negative charge in the structure [18].

Tautomer structures were parameterized by CgenFF (CHARMM-GUI tools, version 3.8). Simulations of the BSA system in the presence of a drug molecule were performed for a system containing 12 tautomers of AN1 and AN3, three repetitions of 100 ns, and one repetition of 300 ns (Figure 9a,b). Seven out of eight trajectories reached conformational stability after about 40 ns. The conformational compactness of BSA in all 100 ns and 300 ns repeats were tabulated and visualized as changes in the RMSD value of the BSA chain as a function of time (Appendix A). It was shown that in all simulations, the AN3 tautomer concentrated much more effectively near the protein molecule, as evidenced by the many times higher value of the radial distribution of the drug from the BSA molecular surface (Figure 9a,b). The influence of the protein’s electrostatic field, assuming the protonation state corresponding to pH = 8.4, is a crucial factor in determining the selective attraction of tautomers towards subdomains, which depends on their polarity. The most intense and frequent maxima of the radial distribution were recorded for subdomains with a net positive charge (IB +2e, IIIA +3e). Less pronounced maxima appeared in the subdomain with a charge closer to neutral (IIA −1e) and the subdomain with the highest net negative charge (IA −12e). The AN3 tautomer was more intensely attracted than AN1 by positively charged amino acids, which results from the greater polarity of the AN3 form. Tautomers were directed to the hydrophilic surface of the protein and interacted electrostatically with positively charged amino acids, i.e., arginine and lysine. During the interaction of BSA with the tautomer, the electrostatic interaction between the negatively charged nitrogen of the tautomer and the protonated amino group of arginine or lysine dominated. The tautomer interaction was polarly stabilized by a hydrogen bond between oxygen in the second or fourth position of the aromatic ring of the tautomer and nitrogen in the outer chain of arginine or lysine.

For both AN1 and AN3, the most stable complexes were recorded in the interdomain region in the center of the protein structure. The identified site was characterized by a high surface density of positive charge and high hydrophilicity (Appendix A). Interaction diagrams were generated for the examined complexes and it was found that they are stabilized by electrostatic interactions and hydrogen bonds of two basic amino acids with a positive charge, i.e., Arg435 and Lys439 (subdomain IIIA, Appendix A). Furthermore, non-covalent attractive forces were recorded between the aromatic ring of the tautomer and the aromatic ring of Tyr451.

The distances of the centers of mass (COMs) of the tautomers from the protein surface were measured as a function of the simulation time, which allowed for a comparison of the number of complexes occurring and their duration. The distances between the centers of mass allowed us to obtain a plateau of function, interpreted as the complexation of the tautomer molecule by BSA. The distances between the plateaus of the COM of the drug and BSA reached between 1.1 and 3.2 nm.

The binding free energy using the gmx_MMPBSA (version 1.6.1) method was determined for the most stable complexes. The energies were calculated from selected trajectory intervals (marked with an black rectangle in Figure 9c). Measurement of the final state energy of the most stable BSA–tautomer complexes allowed for a comparison of the affinity of tautomers to the protein. It was shown that the binding free energy for AN1 and AN3 was −7.79 ± 5.34 kcal/mol and −9.29 ± 4.98 kcal/mol, respectively. The differences in binding free energy between the tautomeric forms are within the margin of error. Using gmx_MMPBSA methods with the PB solvation model (Appendix A) allowed both tautomeric forms to obtain similar energy values. PB methods make it possible to approximate physical effects, i.e., solvent polarizability and ionic effects resulting from the adopted ionic forces in the simulation performed, as well as simplified features of the adopted water model, e.g., TIP3P.

### 2.8. Molecular Docking

The results of the most representative configurations of the twelve docked molecules were visualized by marking the drug locations on the BSA framework model and the VdW surface containing the electrostatic potential distribution. Based on these simulations, the concentration of the AN3 form was observed mainly in the interdomain center with a clear shift towards the IA subdomain (IIIA, IIA, and IA, as demonstrated by MD simulations measuring drug distribution around the protein subdomains (Figure 9a,b)). In the case of the AN1 form, it localized mainly in the IIIA and IIA subdomains. In both cases, more molecules were located at the protein surface than inside it (Figure 10a).

The locations of drug particles in the protein structure were analyzed considering the positions of fluorine atoms on the protein surface. The locations of tautomer molecules greater than 3.2 Å are summarized in a histogram. It was recorded that molecules of lower polarity (AN1) localized with greater frequency in deeper layers from the SASA protein surface, as evidenced by the shift in the histogram shown in Figure 10b. The AN3 form was more often located at the BSA surface than the AN1 form.

Docking tautomers using the multiple-docking method allowed affinity values to be obtained for a drug cluster of a controlled size. For the configuration of twelve tautomers, corresponding to the number of simulated molecules, similar values of the Gibbs free energy for AN1 and AN3 were obtained (−49.81 kcal/mol and −50.42 kcal/mol, respectively). Docking was performed by repeatedly reducing the number of tautomers in the box from twelve molecules to one. In this way, an approximate energy distribution was obtained as a function of the number of tautomers. The obtained energies represent the average binding free energy for each cluster size (top diagram of Appendix A). Each cluster’s average binding free energy was converted into the average binding free energy of one tautomer. The graph showed that the increasing number of tautomers increased the average value of binding free energy, and for twelve ligands, it stabilized at approximately −4 kcal/mol. A similar trend of energy increase was observed for both tautomeric forms (bottom diagram of Appendix A). The largest energy differences between AN1 and AN3 were observed at four tautomers in the docking box. The decreasing value of binding free energy with increasing number of molecules can be explained by the saturation of BSA binding sites, and thus giving the most favorable energy values. For single molecules docked throughout the volume, comparable energy values were obtained between AN1 and AN3 (−5.089 ± 0.230 kcal/mol (Tyr30, Gln32) and −5.176 ± 0.171 kcal/mol (Asp86, Met87, Tyr30), respectively) (Figure 10c). Considering the standard deviation, the results for the two variants of the molecule can be considered to be very similar. Previous studies using AutoDock 4.2.3 showed that BSA complexes with the non-ionized form of 5FU with an energy of −4.39 kcal/mol (Try30, Leu31, Gln32, Gln33, Met87, Lis106, Asp107) [15]. In other work research using Glide software for 5FU, better docking values were obtained: −8.13 kcal/mol (Ser192, Gly195, Tyr149) and −7.00 kcal/mol (Trp134, Glu17) [19]. The differences in the binding free energy of 5FU obtained by previous researchers can be attributed to BSA’s initial conformation and the docking software’s different computational accuracies.

Configurations of twelve docked molecules were selected to determine the spatial distribution of the drug in the protein structure. We examined what percentage of tautomers are located near the surface and what the approximate location of the drug is on the SASA surface of the protein. For this purpose, the EDTSurf algorithm was used, and the fluorine atom was selected as the reference atom. The statistics were based on nine configurations of twelve docked molecules generated by the AutoDockVina program, and the depth of 108 docked molecules for AN1 and AN3 was compared. It was shown that for both tautomers, the proportion of ligands located deeper than 3.2 Å was, on average, 20%, and approximately 80% were ligands fully accessible to SASA (Appendix A). In the MD simulation measuring the distance from the COM of the drug to the COM of the BSA, we also observed short complex durations closer to the protein surface for AN3 (black, red, and blue curves) and AN1 (red curve) Figure 9c.

The docking results of single tautomers were visualized in the form of interaction diagrams, and the locations in the subdomain and the distribution of the electrostatic potential were highlighted. Single tautomers, docked throughout the volume, showed the highest affinity for the IA subdomain (Figure 10c). Lower energy configurations for both tautomeric forms included subdomains IB, IIA, and IIIA (Appendix A). The slight energy differences between the AN1 and AN3 forms and within the group allowed us to conclude that the remaining configurations are equally probable. According to the literature, the non-ionized form of 5FU can be located in the IA and IB subdomains [15].

Because the 5FU molecule is relatively small, it can get into the internal pockets in the interdomain space of albumin without significant problems. But, in addition to the binding sites in the internal pockets, other binding sites are also dedicated to native albumin ligands. It is worth mentioning here that there are binding sites dedicated to fatty acids, which are located on the surface of the protein structure [33]. This would explain the change in the surface properties of the molecule observed in the experimental studies resulting directly from the immobilization of the ligand molecules on the structure’s surface. Simulations for ibuprofen in neutral and deprotonated forms show as many as eight binding sites. Evola et al. defined two individual binding sites as the drug sites (DS1 and DS2 (Sudlow I and II), four positions as the fatty acid binding sites (FA1, FA2, FA5, and FA6), and the two remaining sites are in the upper and lower region of the protein cleft (PCup and PCdown) [34]. These simulations show that charged ibuprofen molecules have a greater affinity for albumin than the uncharged form. Interestingly, in both cases, the spatial arrangement of the binding sites was identical, only the orientation of the molecule took different variants. It should be emphasized that the electrostatic interaction between the charged form of the drug and the polar residues significantly contributes to the binding energy of the ligand. In the case of 5FU tautomers, we observed the same correlations as in the case of the ibuprofen molecule regarding ligand location and binding energy. It should be emphasized that, based on the results of many simulations, the docking of the ligand in different configurations at the same binding site was observed.

## 3. Materials and Methods

### 3.1. Materials

Bovine serum albumin (A0281, CAS: 9048-46-8 fatty acid-free and essentially globulin free, ≥99%, agarose gel electrophoresis) and 5-fluorouracil (F6627, CAS: 51-21-8, ≥99%, HPLC) were purchased from Sigma Aldrich (Saint Louis, MI, USA). All reagents used were of analytical grade. All solutions were prepared by dissolving the powder in distilled and degassed water. The pH of the samples was adjusted using a sodium hydroxide (NaOH) solution. All experiments were carried out at a temperature of 298 K.

### 3.2. Preparing of BSA-5FU Complexes

The measurements were performed for different BSA/5FU molar ratios: 1:10, 1:20, 1:40, and 1:60. The BSA concentration was constant at 0.25 mg/mL (3.8 μM). All complexes were prepared in distilled water, and the pH of all samples was adjusted to pH = 8.4. The BSA/5FU mixtures were stirred in the dark for 1 h. To remove uncomplexed drug molecules, the dialysis of complexes was performed using SnakeSkin™ dialysis tubes (MWCO 3.5 kDa, Thermo Fisher Scientific, Waltham, MA, USA). This process was carried out in distilled water at pH 8.4 for 24 h at room temperature.

### 3.3. UV-Vis Spectroscopy

The UV-vis spectra were measured using a UV-vis spectrophotometer (Thermo Scientific Evolution 201, Waltham, MA, USA) in the wavelength range from 190 nm to 500 nm. UV-vis measurements were conducted to plot the calibration curve of 5FU at pH 8.4 in water (c = 7.7–192 μM) to verify the drug concentration. Spectra of complexes in water at pH = 8.4 were performed in at BSA/5FU molar ratios of 1:10, 1:20, 1:40, and 1:60. A comparative spectrum of BSA was performed for an analogous concentration to that of the complexes (c = 0.25 mg/mL), and a spectrum of the free drug was performed for a concentration corresponding to a BSA/5FU molar ratio of 1:20 (c = 1 × 10^−2^ mg/mL). The spectrum of the same amount of 5-fluorouracil was subtracted from the spectrum of BSA-5FU at a molar ratio of 1:20 to obtain the difference spectra.

### 3.4. Fluorescence Spectroscopy

Fluorescence emission spectra were obtained using a Fluorolog spectrofluorometer (HORIBA Jobin Yvon Ltd., Middlesex, UK) using 1 nm resolution, with both emission and excitation monochromator slits set to 5 nm. Samples in quartz cuvettes, 1 × 1 × 4 cm, with relevant solutions were prepared immediately before the measurements. Cuvettes were covered by parafilm to minimize the air exposure of the samples, which would accelerate the sample degradation and drop the pH value. To determine the binding constants, samples containing 0.5 mL of BSA or HSA with a concentration of 0.33 mg/mL (10 μM) and pH = 8.4 were mixed by adding 0.5 mL of 5FU solution (pH = 8.4) to obtain a final drug concentration in the range of c = 2.6 × 10^−3^–22.0 × 10^−3^ mg/mL (20–160 μM). All fluorescence spectra were measured at two excitation wavelengths of λ_ex_ = 279 nm and λ_ex_ = 296 nm. The emission was recorded between λ_em_ = 300 and 500 nm.

Changes in fluorescence intensity of the considered albumins in the presence of negatively charged 5FU at λ_em_ = 330 nm were used to calculate the BSA-5FU (K_BSA-5FU_) and HSA-5FU (K_HSA-5FU_) binding constants using the following equation [19]:Log[(*F*_0_/*F*)/*F*] = Log*K_a_* + *n* log[*Q*] (1)
where *F*_0_—original fluorescence intensity; *F*—quenched intensity of the fluorophore; *K_a_*—binding constant [M^−1^]; and [*Q*]—molar concentration of the quencher. A plot of log[(*F*_0_/*F*)/*F*] versus log[*Q*] gives a straight line. From the slope of the linear curve, we obtained the value of the binding constant *K_a_*.

### 3.5. Dynamic Light Scattering (DLS)

Dynamic light scattering (DLS) was performed using a Zetasizer Malvern Nano ZS (Malvern Panalytical, Worcestershire, UK) equipped with a 4 mW He–Ne laser operating at a wavelength of 633 nm with a fixed detector angle of 173°. The BSA concentration for all samples was constant at 0.5 mg/mL. The DLS measurements were conducted for BSA/5FU complexes at molar ratios of 1:10, 1:20, 1:40, and 1:60. The hydrodynamic radius (R_H_) of the molecules was calculated using the Stokes–Einstein equation [35]:(2)$$R_{H}=\frac{kT}{6\pi \eta D}$$
where *k*—Boltzmann constant; *T*—absolute temperature; η—viscosity of medium; and *D*—diffusion coefficient.

### 3.6. Electrophoretic Mobility Measurement

Electrophoretic mobility (µ_e_) was determined using a Malvern Nano ZS analyzer (Worcestershire, UK). The measurements for pure BSA and BSA-5FU complexes were made in water with a constant BSA concentration of 1 mg/mL. Measurements were carried out for complexes with BSA/5FU molar ratios of 1:40 and 1:60. Complexes for electrophoretic mobility measurements were mixed for one h and then dialyzed using SnakeSkin™ dialysis tubes (MWCO 3.5 kDa) (Thermo Fisher, Waltham, MA, USA). Dialysis was conducted in deionized water for 24 hours in darkness. All electrophoretic mobility measurements were conducted in the pH range of 2.0–11.0. The pH of solutions was adjusted using sodium hydroxide (NaOH) at a concentration of 0.1 M and hydrochloric acid (HCl) at a concentration of 0.05 M. The zeta potential was related to the electrophoretic mobility via Henry’s equation. The Hückel limit (f(κa) = 1.0) was applied for the calculations.

### 3.7. Quartz Microbalance with Energy Dissipation (QCM-D)

The adsorption process of BSA-5FU complexes on the gold surface of the QCM-D sensor was conducted using the Q-Sense E1 device (Biolin Scientific, Stockholm, Stockholms Lan, Sweden). A constant flow rate of 0.5 mL/s was maintained. Changes in resonance frequency (Δ*F*) and energy dissipation (Δ*D*) were recorded during the experiments and analyzed for the 7th overtone. The QCM-D measurements comprised three phrases: establishing a baseline using distilled water at pH = 8.4 (t = 10 min), adsorption of BSA-5FU complexes at pH = 8.4 (t = 45 min), and rinsing with distilled water at pH = 8.4 (t = 45 min). All solutions intended for adsorption were previously degassed. Due to the low values of dissipation during the measurements (Δ*D* < 1.0 × 10^−6^), the resulting layers can be considered rigid, which made it possible to use the Sauerbrey model to calculate the mass adsorbed on the Au surface [36]. The Sauerbrey model assumes a proportional relationship between the measured chance in resonance frequency (Δ*F*) and the adsorbed mass (*Γ_QCM-D_*) and is expressed by the following equation [37,38]:(3)$$\Gamma_{QCM-D}=-C\frac{\Delta F}{n}$$
where *C*—constant characteristic for quartz crystals; Δ*F*—change in the resonance frequency; and *n*—number of overtones (in the present case, *n* = 7).

### 3.8. Contact Angle Measurements

The hydrophilicity/hydrophobicity of the layers formed by the BSA-5FU complexes on the surface of the QCM-D sensor after each QCM-D measurement was determined using a goniometer (Krüss DSA 100, Hamburg, Germany). The axisymmetric Drop Shape Analysis (ADSA) method obtained the contact angles. The contact angles were determined using the La-Place Young equation [39]:(4)$$cos\theta_{Young}=\frac{\gamma_{SV}-\gamma_{SL}}{\gamma_{LV}}$$
where $\theta_{Young}$—Young’s contact angle; $\gamma_{SV}$—interfacial tension in the solid–gas system; $\gamma_{SL}$—interfacial tension in the solid–liquid system; and $\gamma_{LV}$—interfacial tension in the liquid–gas system.

### 3.9. Circular Dichroism Spectroscopy (CD)

Circular dichroism (CD) measurements were performed using a Jasco J-1500 spectropolarimeter (Tokyo, Japan) with a Peltier-type thermostatic cell holder. Quartz cuvettes with a 1 mm path length were used (Helma Analytics, Müllheim, Germany). The CD spectra of BSA were recorded in the absence and presence of 5FU. The concentration of BSA was kept constate at 0.25 mg/mL. The molar ratios of BSA to 5FU were 1:10, 1:20, 1:40, and 1:60. The spectra of BSA and its complex with drugs were recorded from 185 to 260 nm, with a scan rate of 50 nm/min and data pitch of 0.1 nm, using three scans. All spectra were measured at 298 K and blank-corrected.

### 3.10. MD Simulation and Trajectory Analysis

The BSA structure from the PDB database (3V03) [40] was used in the simulations. The structure of 5FU was downloaded from the DrugBank database. The protein structure was completed with missing atoms using CHARMM-GUI [41] and the BSA protonation corresponding to pH 8.4 conditions was calculated using PDB2PQR. Protein and 5FU tautomer input files were generated using the CHARMM36m force field. Based on the protonated 5FU structure taken from DrugBank, tautomers were constructed by manual deprotonation. The Marvin JS tool (CHARMM-GUI tools) was used to obtain anions with a resultant charge of −1e. The tautomers AN1 and AN3 were deprotonated on the nitrogen of the N1 and N3 positions, respectively. The obtained structures were entered into CgenFF and parameterized, thus obtaining files for simulation and docking. For the most representative simulation, an electrostatic potential distribution was generated using the APBS method and visualized with PyMol [41]. Protein hydrophobicity and hydrophilicity distributions and two-dimensional protein–drug interaction diagrams were also generated using BIOVIA Discovery Studio tools [42]. To estimate the protein–ligand binding affinity, the MMPBSA method was used. BSA-5FU complexes were constructed using the CHARMM-GUI tool. In the simulations of tautomers in the BSA structure, twelve molecules of the AN1 and AN3 tautomers were assumed and simulated using the GROMACS program. The simulations assumed the following conditions: temperature 298.15 K, pressure 1Bar, and water model TIP3P. The system charge was neutralized by adding sodium and chloride counter ions to bring the concentration to 0.01 M. The van der Waals interactions were evaluated using the Lennard–Jones 6–12 potential with a cutoff distance of 12 Å, while the electrostatic interactions were calculated using the Particle Mesh Ewald. Using the GROMACS software tools (version 2022.2), the radial distribution of tautomers interacting with each domain was calculated separately for 100 ns and 300 ns simulations.

### 3.11. Molecular Docking

The EDTSurf program was used to determine the depth of drug localization in the BSA structure. The location of the drug molecules was determined based on the distance between the center of the fluorine atom and the outer solvent-accessible surface area (SASA) with probe radius equal to 1.4 Å. The split of drug localization on the surface of the structure or inside was assumed for a distance of 3.2 Å. The following input parameters were used in the AutoDock Vina program: exhaustiveness of 32, energy threshold of 4.0 kcal/mol, protein and drug ionizations corresponding to pH 8.4 conditions, and ionic strength I = 0.01NaCl. Docking was performed for the total volume of the protein surface. The BSA frame from the MD simulation was used for drug docking. The Gibbs free energy of binding of 5FU tautomer molecules to the protein structure was determined. Multiple docking was performed for a system containing from 1 to 12 ligands, assuming a rigid protein and a rigid drug.

## 4. Conclusions

The work focused on the formation of stable complexes between tautomeric forms of 5FU molecules and bovine serum albumin. The experimental studies demonstrated a binding preference for AN1 and AN3 tautomers in the BSA structure at pH > 8.0. The complexes were studied in solution by fluorescence spectroscopy, UV–vis absorption spectroscopy, circular dichroism (CD) spectroscopy, and dynamic light scattering (DLS). For the first time in the literature, it was shown that anionic forms of 5FU effectively integrate into the inside of the protein structure and on the structure’s surface. The fluorescence spectra confirm that the drug binds effectively to the BSA structure close to tryptophan 212, producing changes in the surrounding microenvironment. The immobilization of drug molecules on the surface of the protein structure does not cause aggregation of the system, although it causes changes in the effective charge of the system. The size of the hydrodynamic diameter increased from 6.3 nm to 7.4 nm. This may result directly from a change in the system’s hydration caused by drug molecules on the surface of the protein structure. The changes within the secondary structure of the protein were slight and local.

A series of MD and docking simulations were performed to determine the preferred drug-binding sites in the protein structure. The binding site of both AN1 and AN3 inside the structure is located mainly in the vicinity of positively charged amino acids found in the interdomain space. At pH 8.4, the interdomain space is positively charged. Under these conditions, the protein has a global negative charge with local positive sites, which may constitute potential binding sites for negatively charged molecules. A high density of surface positive charge and high hydrophilicity characterize the identified active site. For single molecules docked throughout the volume, similar values of the Gibbs free energy were obtained for AN1 and AN3 (ΔG = −5.089 kcal/mol and ΔG = −5.176 kcal/mol, respectively). In simulations that included 12 drug molecules in the system, 20% were located inside and 80% on the protein’s surface. Changes in the physicochemical property of a protein upon ligand addition have also been considered as indirect evidence of ligand–protein interactions. The potential immobilization of negatively charged 5FU tautomers was confirmed by changes in the zeta potential of the complexes, the adsorption efficiency on the Au surface, and the contact angles of the adsorbed layers.

It should be emphasized that determining the physicochemical factors that play an essential role in the interaction of 5-FU molecules with BSA is significant progress in understanding the drug’s mechanism of action in cancer therapy and the path for selecting new drug candidates.

## Figures and Tables

**Figure 1 ijms-25-00037-f001:**
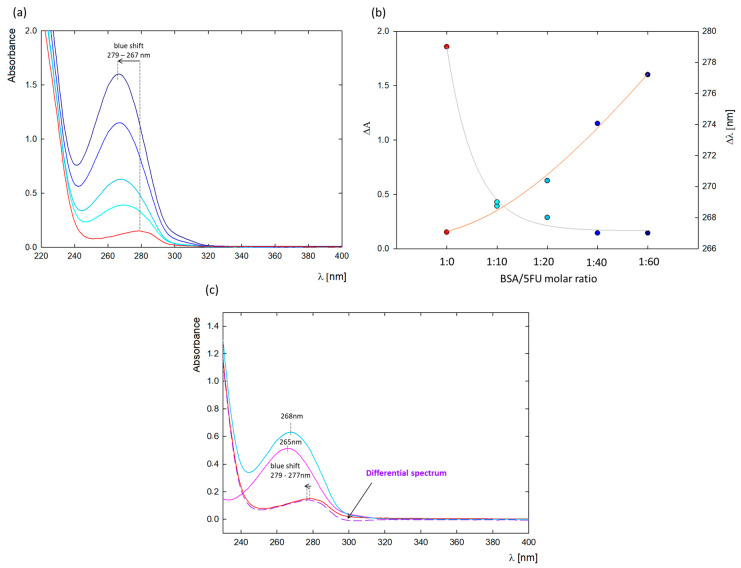
(**a**) UV-vis spectra of BSA-5FU complexes depending on the 5FU concentration (1:10 (cyan), 1:20 (light blue), 1:40 (blue), and 1:60 (dark blue)) compared to pure BSA at c = 0.25 mg/mL (red curve); (**b**) effect of the amount of added 5FU on the absorbance value (orange curve) and the peak maximum position (gray curve); (**c**) difference spectrum (purple dashed curve) compared to BSA spectrum (red curve), 5FU at the same concentration as in the complex (pink curve), and BSA-5FU 1:20 (blue curve).

**Figure 2 ijms-25-00037-f002:**
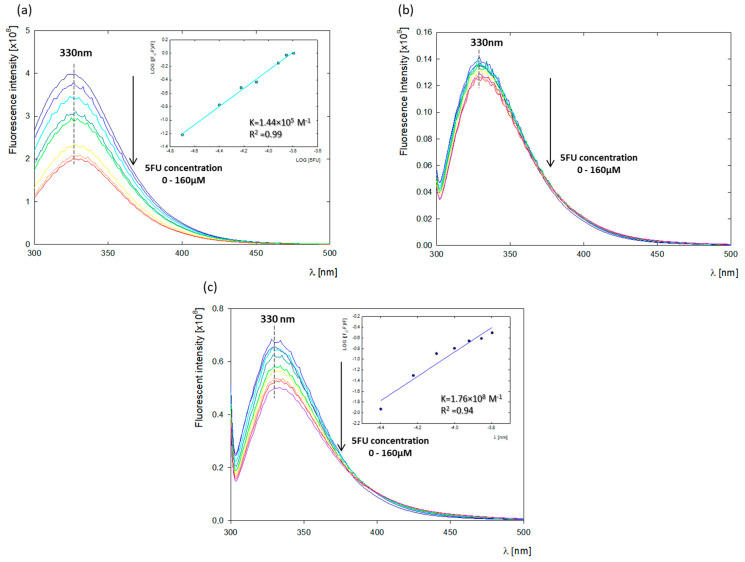
Fluorescence quenching titrations of albumins (c = 10 μM) with increasing 5FU concentrations (20–160 μM) studied in water at pH 8.4 and the plots of log[(*F*_0_ − *F*)/*F*] versus log[*5FU*]; (**a**) BSA titration by 5FU upon excitation at 279 nm; (**b**) BSA titration by 5FU upon excitation at 296 nm; (**c**) HSA titration by 5FU upon excitation at 296 nm.

**Figure 3 ijms-25-00037-f003:**
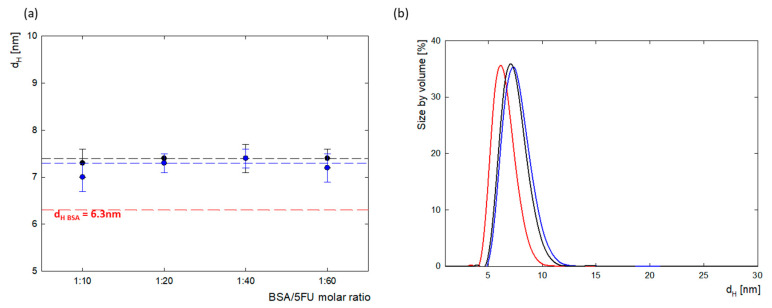
(**a**) The comparison of the hydrodynamic diameter (d_H_) calculated from maximum in the scattering peak of BSA-5FU complexes depending on BSA/5FU molar ratio, both before dialysis (black) and after dialysis (blue) compared to pure BSA (red); (**b**) particle size distributions by volume of BSA-5FU complexes at a molar ratio of 1:20 before dialysis (black) and after dialysis (blue) compared to pure BSA (red) (c_BSA_ = 0.5 mg/mL, H_2_O).

**Figure 4 ijms-25-00037-f004:**
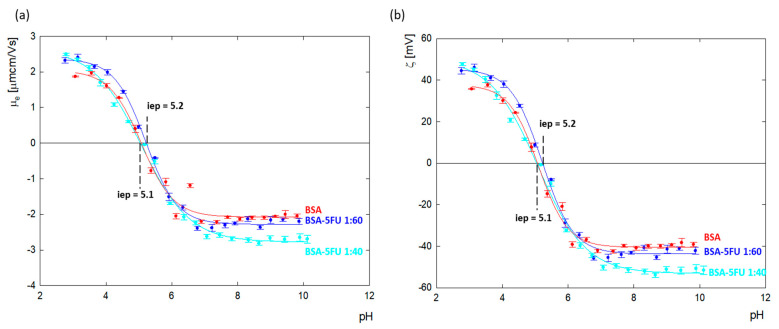
(**a**) Electrophoretic mobility and (**b**) zeta potential as a function of pH for an aqueous solution of BSA (c_BSA_= 1.0 mg/mL) and its complexes with 5FU with ratios of 1:40 and 1:60 after dialysis.

**Figure 5 ijms-25-00037-f005:**
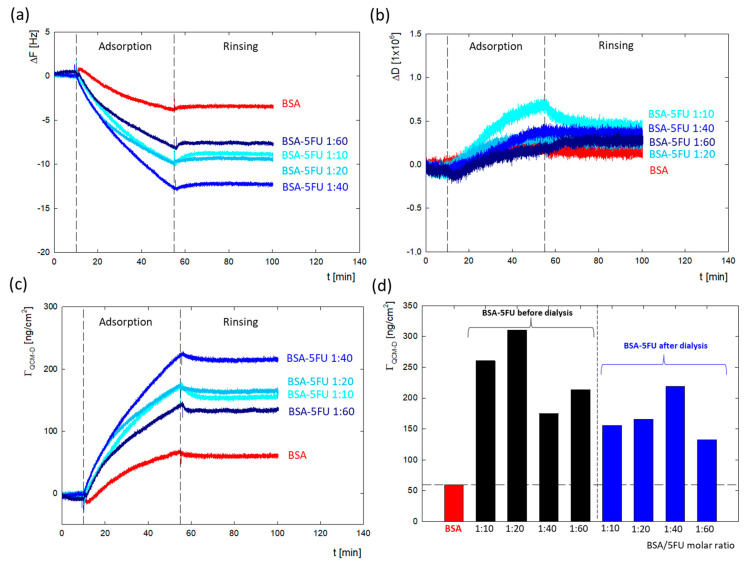
Comparison of adsorption of BSA and BSA-5FU complexes on Au surface monitored using QCM-D depending on 5FU concentration and dialysis stage with (**a**) changes in resonance frequency (Δ*F*); (**b**) changes in dissipation energy (Δ*D*); (**c**) changes in adsorbed mass (Γ_QCM-D_). (**d**) Summary of the mass of adsorbed BSA and BSA-5FU complexes (c_BSA_ = 0.25 mg/mL, H_2_O).

**Figure 6 ijms-25-00037-f006:**
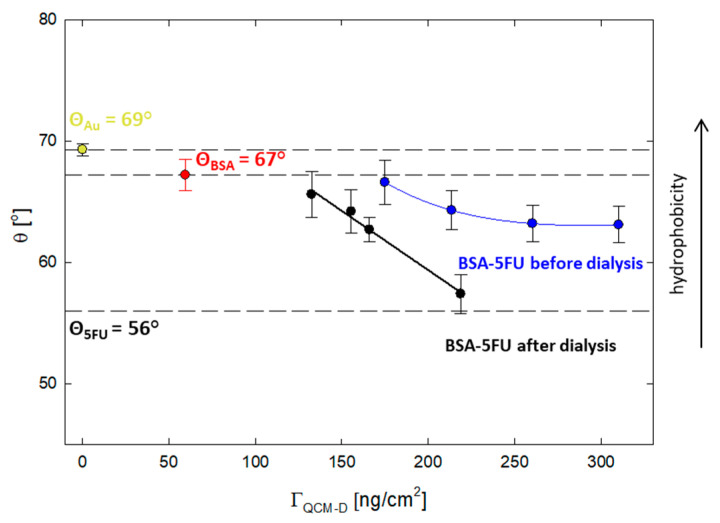
The correlation between the acquired contact angles and the mass adsorbed on the Au surface for BSA-5FU complexes at different molar ratios, both before and after dialysis.

**Figure 7 ijms-25-00037-f007:**
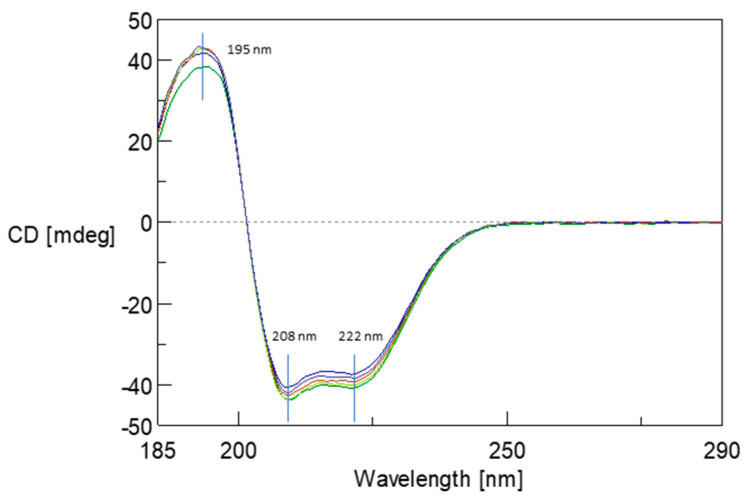
The circular dichroism (CD) spectra of BSA (green line) and BSA upon interaction with 5FU molecules at a ratios of 1:10 (light green line) to 1:60 (navy line) at pH 8.4.

**Figure 8 ijms-25-00037-f008:**
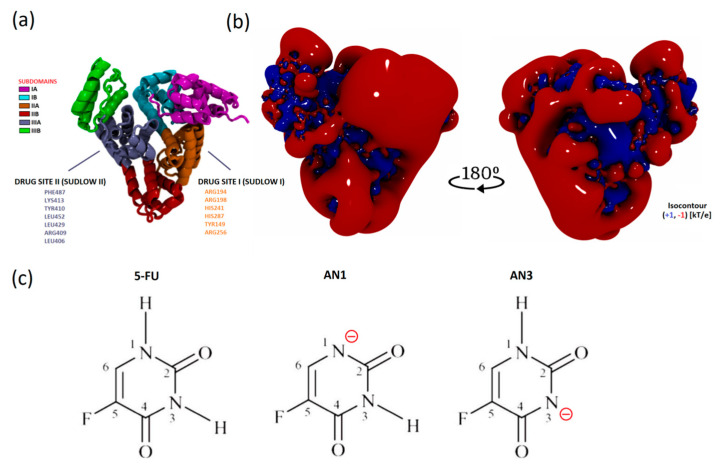
Visualization of the spatial structures of the molecules simulated: (**a**) the secondary structure of BSA and subdomains based on a representative simulation frame; (**b**) the electrostatic potential contour of BSA in the protonation state at pH 8.4 (representative conformation after MD simulation, intended for docking); (**c**) structural formulae of 5FU and its tautomers AN1 and AN3 with nitrogen deprotonation sites labeled.

**Figure 9 ijms-25-00037-f009:**
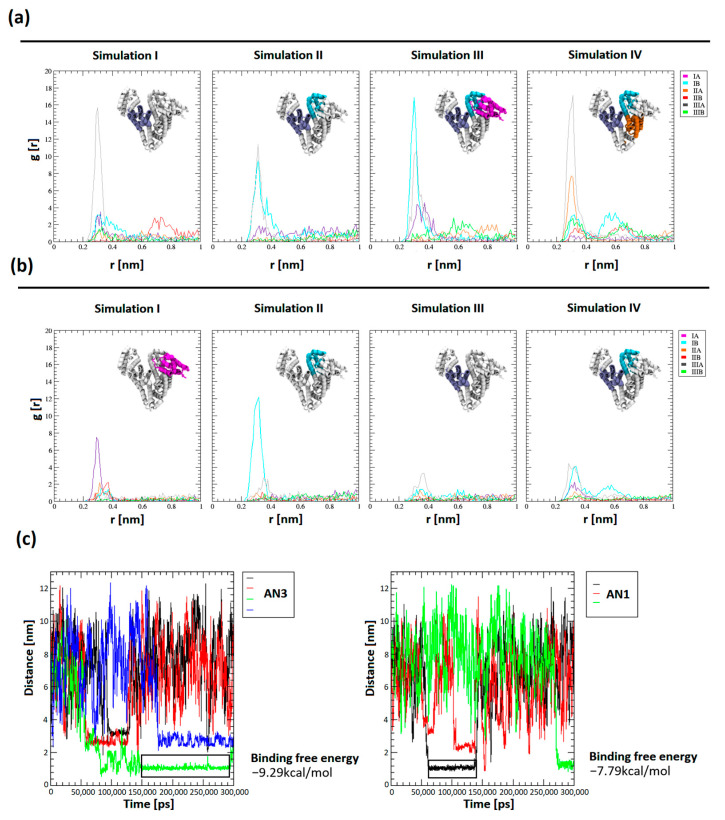
Comparison of affinity of tautomers to BSA subdomains after simulations. (**a**,**b**) Spatial distribution of tautomers around BSA subdomains (all 100 ns and 300 ns repetitions); (**c**) durations of protein–tautomer complexes as determined by the distance between the COM of the drug and BSA (performed for 300 ns trajectories). Representative ranges of trajectories designed to measure the free energy of binding (gmx_MMPBSA) are marked with a black rectangle.

**Figure 10 ijms-25-00037-f010:**
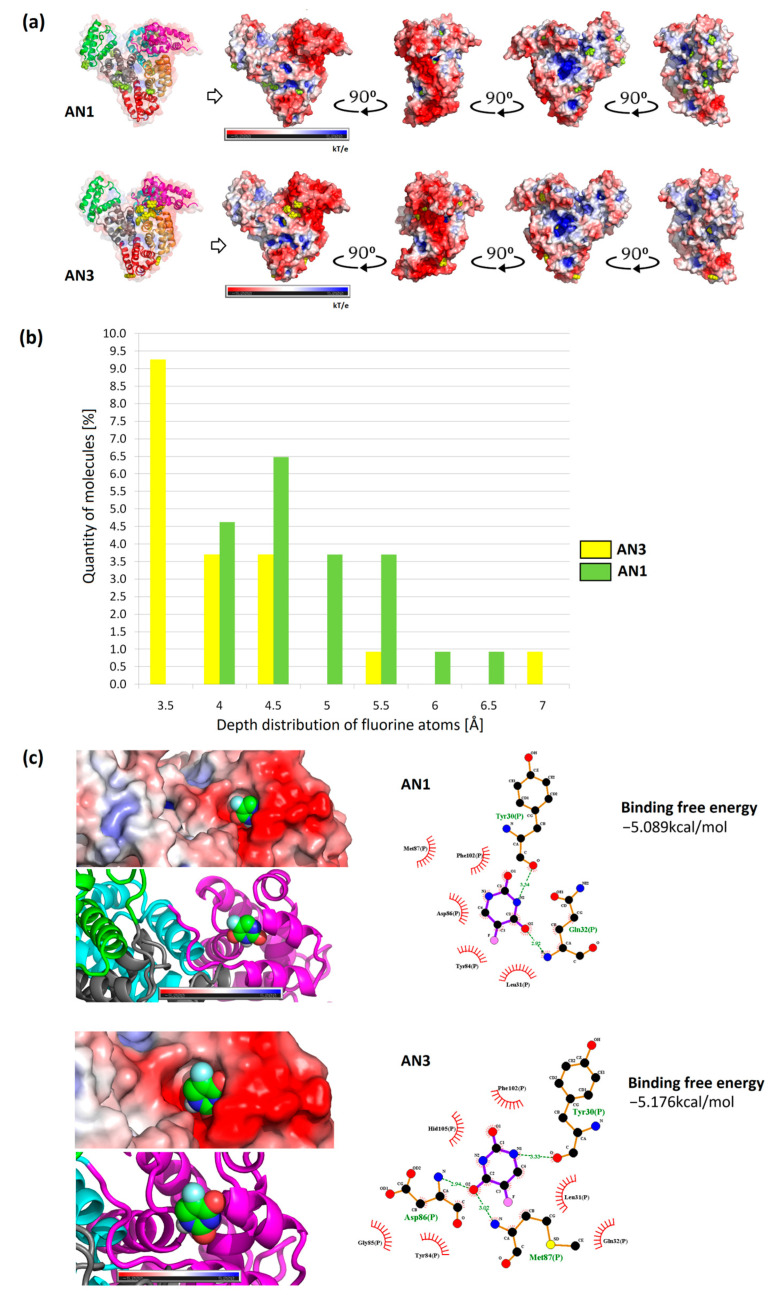
Summary of AutoDock Vina docking results and tautomer depth measurements (EDTSurf). (**a**) The most representative configurations from the docking of the twelve AN1 and AN3 molecules; (**b**) comparison of the depth distribution of fluorine atoms for the twelve tautomer complexes; (**c**) interaction diagram for the lowest docking free energies of the single tautomers AN1 and AN3.

## Data Availability

Data are contained within the article and Appendix A.

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
