# Peer review of "Bovine Serum Albumin as a Platform for Designing Biologically Active Nanocarriers—Experimental and Computational Studies"

_ijms, 2023, doi:10.3390/ijms25010037_

Round 1

Reviewer 1 Report

Comments and Suggestions for Authors

The article “BSA as a platform for designing biologically active nanocarriers – experimental and computational studies” is devoted to a current topic – the development of targeted drug delivery systems.

The manuscript involves determining the effectiveness of BSA as a potential carrier of 5-fluorouracil (5-FU) as an anticancer drug. The research was carried out to estimate the physicochemical properties of the system using complementary measurement techniques. UV-vis absorption, fluorescence quenching, zeta potential, QCM-D, and CD spectroscopic studies were performed. The experimental research was supported by molecular dynamics (MD) simulations and molecular docking.

The article is well structured, the methods are described in sufficient detail, which allows us not to doubt the results obtained by the authors and the correctness of the conclusions.

I just have a few technical comments:

1. There are many strange word wrap in the text of the article, for example, Line 11 – “lig-ands” (better “li-gands”), Line 12 – “ther-apeutic” (better “thera-peutic”). Such word wrap should be double-checked throughout the text of the article.

2. Before references “…[1]”, “…[2]”, “…[3]”, “…[6]” in the text authors should put spaces, in particular (but not only): lines 33, 35, 38, 44 and further in the text.

In addition, instead of “…[3][4]” and “…[6][7]”, format it as “…[3, 4]” and “…[6, 7]”. Likewise throughout the entire text of the article.

3. The title of the Section “2.4 Electrophoretic mobility” should be moved to page 6, where the text of this section begins.

4. In Section “3. Materials and methods" authors should enter the following subsections: "3.1. Materials", "3.2. Preparation of BSA-5FU complexes", "3.3. UV-vis spectroscopy", etc.

5. After the subsections “Molecular docking” authors should remove the blank line.

6. In Reference 42, the capital letters from “D. BIOVIA” should be removed.

Author Response

Point-by-point response to reviewer’s comments

Changes made into the manuscript are marked

We would like to thank the Reviewers for their constructive comments, and the attention paid to our manuscript. Below are the detailed answers to the referees’ reports:

Reviewer 1: The article “BSA as a platform for designing biologically active nanocarriers – experimental and computational studies” is devoted to a current topic – the development of targeted drug delivery systems.The manuscript involves determining the effectiveness of BSA as a potential carrier of 5-fluorouracil (5-FU) as an anticancer drug. The research was carried out to estimate the physicochemical properties of the system using complementary measurement techniques. UV-vis absorption, fluorescence quenching, zeta potential, QCM-D, and CD spectroscopic studies were performed. The experimental research was supported by molecular dynamics (MD) simulations and molecular docking.The article is well structured, the methods are described in sufficient detail, which allows us not to doubt the results obtained by the authors and the correctness of the conclusions.I just have a few technical comments:

  1. There are many strange word wrap in the text of the article, for example, Line 11 – “lig-ands” (better “li-gands”), Line 12 – “ther-apeutic” (better “thera-peutic”). Such word wrap should be double-checked throughout the text of the article.
  2. Before references “…[1]”, “…[2]”, “…[3]”, “…[6]” in the text authors should put spaces, in particular (but not only): lines 33, 35, 38, 44 and further in the text.

In addition, instead of “…[3][4]” and “…[6][7]”, format it as “…[3, 4]” and “…[6, 7]”. Likewise throughout the entire text of the article.

  1. The title of the Section “2.4 Electrophoretic mobility” should be moved to page 6, where the text of this section begins.
  2. In Section “3. Materials and methods" authors should enter the following subsections: "3.1. Materials", "3.2. Preparation of BSA-5FU complexes", "3.3. UV-vis spectroscopy", etc.
  3. After the subsections “Molecular docking” authors should remove the blank line.
  4. In Reference 42, the capital letters from “D. BIOVIA” should be removed.

Author replies to comments 2-6: Thank you for checking the manuscript in detail. All formatting deficiencies have been corrected.

Reviewer 2 Report

Comments and Suggestions for Authors

Authors proposed a paper entitled “BSA as a platform for designing biologically active nanocarriers experimental and computational studies” for the publication in International Journal of Molecular Sciences

The paper has a good scientific soundness.

The use of English is quite good

The paper deserves to be published after performing some revisions.

Therefore, here is the list of my issues:

Line 33. “One of the most promising DDS strategies in-33 volves the use of various types of nanoparticles.” I would not say directly that the strategy involves the use of nanoparticles. I would go deeply into this concept, expressing that the real strategy is to entrap drugs or active molecules into nanoparticles. Moreover, the difference among nano and microparticles stands in their bio-absorbability.

Line 36. “nanosized materials ranging in size from 1 to 100 nm for biomedical purposes” also on this sentence, I may have some issues. In particular, nanoparticles is a concept that it is quite difficult to just relate to mean dimensions. Nanometric dimensions for drug delivery systems generally may be considere under 300 nm. Larger dimensions start to be considered sub-micrometric. However, the range from 1 to 100 nm tends to be too restrictive, in my opinion.

Line 38. Please unify reference 3 with Nr. 4. Same observation about Ref. Nr. 6 and Nr. 7. Same in Line 66.

Line 98 “The simulations 98 confirmed the location of the 5FU tautomer” this seems to be written as it were a result. In this final part of the introduction, a paragraph containing the goals and intentions of this paper should be reported clearly.

Line 105. “The most significant changes occur at λ=280” are there any confirmation in the literature of this?

Line 109. “BSA alone” I would substitute with “native BSA”.

Figure 1. Here the figure 1b is embedded in figure 1a. I suggest extract figure 1b and show it regularly between figure 1a and figure 1c.

Figure 1c. I would cut the y-axis of this figure at 1.5.

Line 120. Please add a space here.

Line 139. This equation should be reported in Methods section. Please, define all the variables, including significance and eventual unit of measure, where they are not dimensionless.

Figure 3b. What would change by using linear x-axis scale?

Line 243. “(minima) at 208 nm” please check “minima”

Line 261. “we conclude” I would not use personal forms, if possible.

Images of Figure 9a and 9a are already compositions of 4 sub-figures each. It is not possible to see details correctly. I would suggest expanding them, creating figures a,b,c,d.

Line 420. “HPLC)were pur” add a space where missing.

Please, add also CAS numbers to reagents reported in this section.

Line 427. “at c=0.25 mg/mL” here the “c=” is not necessary.

Line 434. Manufacturer society and city of the manufacturer are missing.

Line 458. Concerning this instrument, same observation as above

Please, check if equations need to be numbered.

Line 516. “database (3V03)[40]was used” please add missing spaces.

Line 541. Please check the format and spaces of this paragraph.

Comments on the Quality of English Language

A quite good use of English

Author Response

Point-by-point response to reviewer’s comments

Changes made into the manuscript are marked

We would like to thank the Reviewers for their constructive comments, and the attention paid to our manuscript. Below are the detailed answers to the referees’ reports:

Reviewer 2: Authors proposed a paper entitled “BSA as a platform for designing biologically active nanocarriers experimental and computational studies” for the publication in International Journal of Molecular Sciences. The paper has a good scientific soundness.The use of English is quite good. The paper deserves to be published after performing some revisions.Therefore, here is the list of my issues:

  1. Line 33. “One of the most promising DDS strategies in-33 volves the use of various types of nanoparticles.” I would not say directly that the strategy involves the use of nanoparticles. I would go deeply into this concept, expressing that the real strategy is to entrap drugs or active molecules into nanoparticles. Moreover, the difference among nano and microparticles stands in their bio-absorbability.

Author reply: The introduction has been revised and supplemented with information justifying the use of nanoparticles in drug delivery systems. In addition, the difference between the properties of nanoparticles compared to microscale particles has been highlighted:

Drug delivery systems (DDS) aim to improve the effects of drugs by extending drug release times, improving biodistribution, increasing the biocompatibility of the carrier, and mitigating side effects of treatment [1]. Drug delivery systems take advantage of the unique properties of nanometer-sized particles to improve the biodistribution and pharmacokinetics of active substances [2-3]. Although it has been accepted that nanoparticles should be smaller than 150 nm, it is increasingly reported that due to the discontinuity of the cancer tumor epithelium, the size requirements for nanoparticles developed for cancer treatment are 70 to 200 nm [4]. Nanoparticles, due to their size, are more efficiently internalized into cells than larger microparticles, making them effective transport and delivery systems [5]. The attractiveness of nanoparticles is based on their relatively high ratio of functional surface area to mass, making them able to efficiently bind and carry bioactive compounds [6]. Nanosized objects are designed at the atomic or molecular level and exhibit unique chemical, magnetic, and biological properties [7,8].

  1. Line 36. “nanosized materials ranging in size from 1 to 100 nm for biomedical purposes” also on this sentence, I may have some issues. In particular, nanoparticles is a concept that it is quite difficult to just relate to mean dimensions. Nanometric dimensions for drug delivery systems generally may be considere under 300 nm. Larger dimensions start to be considered sub-micrometric. However, the range from 1 to 100 nm tends to be too restrictive, in my opinion.

Author reply: We agree that after delving into the literature data, the size range of 1-100 nm is too restrictive. Especially considering cancer therapy, where endothelial discontinuity results in pores within the tumor microenvironment having sizes as large as 780 nm. A particle size of 70-200nm is reported as the optimal size for anti-cancer therapies. Accordingly, the introduction was modified as follows:

The sentence suggesting that only nanoparticles between 1 - 100nm in size are used in biomedicine has been removed: “Nanomedicine involves the design of nanosized materials ranging in size from 1 to 100 nm for biomedical purposes.”

New information about the optimal size of nanoparticles for drug transport has been added, according to literature data: “Although it has been accepted that nanoparticles should be smaller than 150 nm, it is increasingly reported that due to the discontinuity of the cancer tumor epithelium, the size requirements for nanoparticles developed for cancer treatment are 70 to 200 nm [4].”

  1. Line 38. Please unify reference 3 with Nr. 4. Same observation about Ref. Nr. 6 and Nr. 7. Same in Line 66.

Author reply: According to the note, references throughout the manuscript have been unified.

  1. Line 98 “The simulations confirmed the location of the 5FU tautomer” this seems to be written as it were a result. In this final part of the introduction, a paragraph containing the goals and intentions of this paper should be reported clearly.

Author reply: In the theoretical part of this paragraph, words more appropriate to theoretical considerations are used.

"The simulations confirmed the location of the 5FU tautomer molecules both inside the structure and on the surface of the system"

Changed to:

"The simulations approximated the distribution of 5FU tautomer molecules both inside the structure and on the surface of the system"

"Thus, the existence of more active sites than expected for protein systems was confirmed"

Changed to:

"Thus, the existence of more active sites than expected for protein systems was proposed

  1. Line 105. “The most significant changes occur at λ=280” are there any confirmation in the literature of this?

Author reply: Specifically, it is the absorption at λ=279nm. By mistake, the wavelength λ=280 nm was given instead of λ=279nm, which was clarified in the manuscript. The sources used to provide this information were also included. They come from the work of Ali et al., where the researchers also use spectroscopic methods to analyze interactions in the BSA-drug system [M. S. Ali et al., J. Biomol. Struct. Dyn., 2022, 40(19), 9144–9157]. Absorption at λ=279 nm represents π->π* transitions in aromatic amino acids such as tryptophan (Trp), tyrosine (Tyr) and phenylalanine (Phe). Ligand interactions with these amino acids are visible on the UV-vis spectrum as changes within the absorption λ=279 nm [M. S. Ali et al., J. Biomol. Struct. Dyn., 2022, 40(19), 9144–9157, H. A. Shereef et al.,Appl. Organomet. Chem., 2022, 36(4), 1–14].

Publications that referred to this information were inserted in the following sentence “Thus, the effect of ligands on changes in their microenvironment can be observed by UV-visible spectroscopy [21, 22].” For clarity, they are also given here: “The most significant changes occur at λ=279 nm, where absorption comes from the three aromatic amino acids in the protein structure (tryptophan, tyrosine, and phenylalanine) [21,22]. Thus, the effect of ligands on changes in their microenvironment can be observed by UV-visible spectroscopy [21, 22]”

  1. Line 109. “BSA alone” I would substitute with “native BSA”.

Author reply: Thank you for your detailed comments. The sentence has been corrected

  1. Figure 1. Here the figure 1b is embedded in figure 1a. I suggest extract figure 1b and show it regularly between figure 1a and figure 1c.

Author reply: As suggested, the figures have been separated.

  1. Figure 1c. I would cut the y-axis of this figure at 1.5.

Author reply: As suggested, the scale has been changed.

  1. Line 120. Please add a space here.

Author reply: Thank you for your detailed comments. The space has been added

10.Line 139. This equation should be reported in Methods section. Please, define all the variables, including significance and eventual unit of measure, where they are not dimensionless.

Author reply: The equation has been moved to subsection 3.4 Fluorescence spectroscopy found in section 3. Materials and Methods. As suggested, all variables have been explained as follows:

“Changes in fluorescence intensity of considered albumins in the presence of negatively charged 5FU at λem=330nm have contributed to the determining BSA-5FU (KBSA-5FU) and HSA-5FU (KHSA-5FU) binding constants, which were calculated using the following equation [19]:

Log [(F0/F)/F] = LogKa + n log[Q]

Where F0 - original fluorescence intensity, F – quenched intensity of the fluorophore, Ka – binding constant [M-1] and [Q] is the molar concentration of the quencher. A plot of log[(F0/F)/F] versus log[Q] gives a straight line. From the slope of the linear curve, we obtained the value of the binding constant Ka.”

11.Figure 3b. What would change by using linear x-axis scale?

Author reply: We changed x-axis scale to linear in Figure 3b. Due to the poor visibility of the distributions in the 1-100nm range, the magnitude range was also changed and the distributions were shown in the 0-30nm size range.

  1. Line 243. “(minima) at 208 nm” please check “minima”

Author reply: Thank you for your detailed comments. The sentence has been corrected

  1. Line 261. “we conclude” I would not use personal forms, if possible.

Author reply: Thank you for your detailed comments. The sentence has been corrected

  1. Images of Figure 9a and 9a are already compositions of 4 sub-figures each. It is not possible to see details correctly. I would suggest expanding them, creating figures a,b,c,d.

Author reply: Figure 9, like Figure 10, was reduced to a-c. Part of the figure went into supplementary material, allowing the graphs to be enlarged. The colors of the curves in the graphs were darkened to increase contrast.

  1. Line 420. “HPLC)were pur” add a space where missing.

Please, add also CAS numbers to reagents reported in this section.

Author reply: Thank you for your detailed comments. The space has been added, and the CAS numbers of the reagents are included as follows:

“Reagents Bovine Serum Albumin (A0281, CAS: 9048-46-8, fatty acid-free and essentially globulin free, ≥ 99%, agarose gel electrophoresis) and 5-Fluorouracil (F6627, CAS: 51-21-8, ≥ 99%,HPLC) were purchased from Sigma Aldrich (Saint Louis, Missouri, USA).”

16.Line 427. “at c=0.25 mg/mL” here the “c=” is not necessary.

Author reply: We agree, the "c=" symbol has been removed.

17.Line 434. Manufacturer society and city of the manufacturer are missing.

Author reply: Thank you for your detailed comments. The sentences has been corrected.

18.Line 458. Concerning this instrument, same observation as above

Author replies to comments 17-18: Information regarding Manufacturer society and city of the manufacturer has been updated as follows:

“The UV-Vis spectra were measured using a UV-Vis spectrophotometer (Thermo Scientific Evolution 201, Waltham, Massachusetts, U.S.) in the wavelength range from 190 nm to 500 nm.”

“Dynamic light scattering (DLS) was performed using a Zetasizer Malvern Nano ZS (Malvern Panalytical, Worcestershire, UK) equipped with a 4 mW He–Ne laser operating at a wavelength of 633 nm with a fixed detector angle of 173°.”

  1. Please, check if equations need to be numbered.

Author reply: The text has been corrected, in accordance with the journal's rules, the equations should be numbered

  1. Line 516. “database (3V03)[40]was used” please add missing spaces.

Author reply: The missing space has been included.

21.Line 541. Please check the format and spaces of this paragraph.

Author reply: Unnecessary space has been removed from this paragraph.

Round 2

Reviewer 2 Report

Comments and Suggestions for Authors

Authors provided a revised version of their paper. Authors responded point by point to my issues of the previous revision round.

The introduction was enriched as requested, with more descriptions.

All the additional information was added to the specific sections, where required

The paper deserves to be published in the present form.